# Thresholds of glycemia, insulin therapy, and risk for severe retinopathy in premature infants: A cohort study

Elsa Kermorvant-Duchemin[1,2,3]*, Guylène Le Meur[4], Frank Plaisant[5], Laetitia Marchand-Martin[6], Cyril Flamant[7,8], Raphaël Porcher[3,6], Alexandre Lapillonne[1,3], Sylvain Chemtob[9], Olivier Claris[5,10], Pierre-Yves Ancel[3,6], Jean-Christophe Rozé[7,8]

1 AP-HP, Necker-Enfants Malades University Hospital, Department of Neonatal Medicine, Paris, France, 2 INSERM (UMRS1138), Cordeliers Research Center, Paris, France, 3 Université de Paris, Paris, France, 4 Department of Ophthalmology, Nantes University Hospital, Nantes, France, 5 Department of Neonatal Medicine, Hôpital Femme Mère Enfant, Hospices Civils de Lyon, Bron, France, 6 Obstetrical, Perinatal, and Pediatric Epidemiology Team, Centre of Research in Epidemiology and Statistics, INSERM (UMR1153), Paris, France, 7 Department of Neonatal Medicine, Nantes University Hospital, Nantes, France, 8 INRA (UMR1280), Physiologie des Adaptations Nutritionnelles, IMAD, Centre de Recherche en Nutrition Humaine Ouest, Nantes, France, 9 Department of Pediatrics, Ophthalmology and Pharmacology, Centre Hospitalier Universitaire Sainte-Justine Research Center, Montréal, Québec, Canada, 10 Lyon University, EA, Lyon, France

* elsa.kermorvant@aphp.fr

## Abstract

### Background

Hyperglycemia in preterm infants may be associated with severe retinopathy of prematurity (ROP) and other morbidities. However, it is uncertain which concentration of blood glucose is associated with increased risk of tissue damage, with little consensus on the cutoff level to treat hyperglycemia. The objective of our study was to examine the association between hyperglycemia and severe ROP in premature infants.

### Methods and findings

In 2 independent, monocentric cohorts of preterm infants born at <30 weeks' gestation (Nantes University Hospital, 2006–2016, primary, and Lyon-HFME University Hospital, 2009–2017, validation), we first analyzed the association between severe (stage 3 or higher) ROP and 2 markers of glucose exposure between birth and day 21—maximum value of glycemia ($MaxGly_{1-21}$) and mean of daily maximum values of glycemia ($MeanMaxGly_{1-21}$)—using logistic regression models. In both the primary ($n = 863$ infants, mean gestational age 27.5 ± 1.4 weeks, boys 52.5%; 38 with severe ROP; 54,083 glucose measurements) and the validation cohort ($n = 316$ infants, mean gestational age 27.4 ± 1.4 weeks, boys 51.3%), $MaxGly_{1-21}$ and $MeanMaxGly_{1-21}$ were significantly associated with an increased risk of severe ROP: odds ratio (OR) 1.21 (95% CI 1.14–1.27, $p < 0.001$) and OR 1.70 (95% CI 1.48–1.94, $p < 0.001$), respectively, in the primary cohort and OR 1.17 (95% CI 1.05–1.32, $p = 0.008$) and OR 1.53 (95% CI 1.20–1.95, $p < 0.001$), respectively, in the validation cohort. These associations remained significant after adjustment for confounders in both cohorts. Second, we identified optimal cutoff values of duration of exposure

**Data Availability Statement:** Despite anonymisation, the data from the primary and the validation cohorts used for the analyses contains potentially identifying patient information, in particular in the validation cohort where the small number of ROP combined with gestational age may allow patient identification. In compliance with the European General Data Protection Regulation, these data cannot be shared in a public repository. According to the rules imposed by the INSERM (Institut National de la Santé et de la Recherche Médicale), sponsor of the EPIPAGE-2 study, access to the EPIPAGE-2 data is subject to authorisation by the cohort Data Access Committee, in accordance with the European General Data Protection Regulation and the new law for modernisation of the French Public Health System voted in 2016. The data underlying the results presented in the study are all available from Valerie Benhammou (valerie.benhammou@inserm.fr) on request from researchers meeting the criteria for access to confidential data. Contact information: Valérie Benhammou, INSERM U1153 - Equipe EPOPé, Bâtiment Recherche - Hôpital Tenon, 4, rue de la Chine 75020 Paris, France, valerie.benhammou@inserm.fr.

**Funding:** The LIFT cohort (PI: JCR) is supported by grants from the Regional Health Agency of Pays de la Loire (https://www.pays-de-la-loire.ars.sante.fr). The EPIPAGE-2 cohort (PI: PYA) is supported by the French Institute of Public Health Research/Institute of Public Health (https://www.iresp.net) and its partners the French Health Ministry (https://solidarites-sante.gouv.fr), the National Institute of Health and Medical Research (https://www.inserm.fr), the National Institute of Cancer (https://www.e-cancer.fr), and the National Solidarity Fund for Autonomy (https://www.cnsa.fr), and a grant ANR-11-EQPX-0038 from the National Research Agency through the French Equipex Program of Investments in the Future (https://anr.fr). The funders had no role in study design, data collection and analysis, decision to publish, or preparation of the manuscript.

**Competing interests:** The authors have declared that no competing interests exist.

**Abbreviations:** IPTW, inverse probability of treatment weighting; MaxGly, maximum glycemia; $MaxGly_{1-21}$, maximum value of glycemia between birth and day 21; $MeanMaxGly_{1-21}$, mean of daily maximum values of glycemia between birth and day 21; NICU, neonatal intensive care unit; OR, odds ratio; RCT, randomized controlled trial; ROC, receiver operating characteristic; ROP, retinopathy of prematurity.

above each concentration of glycemia between 7 and 13 mmol/l using receiver operating characteristic curve analyses in the primary cohort. Optimal cutoff values for predicting stage 3 or higher ROP were 9, 6, 5, 3, 2, 2, and 1 days above a glycemic threshold of 7, 8, 9, 10, 11, 12, and 13 mmol/l, respectively. Severe exposure was defined as at least 1 exposure above 1 of the optimal cutoffs. Severe ROP was significantly more common in infants with severe exposure in both the primary (10.9% versus 0.6%, $p < 0.001$) and validation (5.2% versus 0.9%, $p = 0.030$) cohorts. Finally, we analyzed the association between insulin therapy and severe ROP in a national population-based prospectively recruited cohort (EPIPAGE-2, 2011, $n = 1,441$, mean gestational age 27.3 ± 1.4, boys 52.5%) using propensity score weighting. Insulin use was significantly associated with severe ROP in overall cohort crude analyses (OR 2.51 [95% CI 1.13–5.58], $p = 0.024$). Adjustment for inverse propensity score (gestational age, sex, birth weight percentile, multiple birth, spontaneous preterm birth, main pregnancy complications, surfactant therapy, duration of oxygen exposure between birth and day 28, digestive state at day 7, caloric intake at day 7, and highest glycemia during the first week) and duration of oxygen therapy had a large but not significant effect on the association between insulin treatment and severe ROP (OR 0.40 [95% CI 0.13–1.24], $p = 0.106$). Limitations of this study include its observational nature and, despite the large number of patients included compared to earlier similar studies, the lack of power to analyze the association between insulin use and retinopathy.

## Conclusions

In this study, we observed that exposure to high glucose concentration is an independent risk factor for severe ROP, and we identified cutoff levels that are significantly associated with increased risk. The clinical impact of avoiding exceeding these thresholds to prevent ROP deserves further evaluation.

## Author summary

### Why was this study done?

- Hyperglycemia, i.e., elevated blood glucose, is common in preterm babies, due to the immaturity of glucose regulation mechanisms; it is often treated with insulin infusion, based on weak evidence.

- A number of studies, with heterogeneity in both setting and design, have suggested that hyperglycemia may be associated with increased risk of morbidity in premature infants, especially severe retinopathy of prematurity, a condition characterized by an abnormal development of the retinal vessels that can lead to blindness.

- However, current published evidence does not rule out that hyperglycemia may only be a marker of severity of illness and not an independent risk factor for retinopathy of prematurity because not all potential confounding factors were taken into account.

- It is also uncertain which concentration of blood glucose could be associated with tissue damage in preterm infants, with little consensus on the cutoff level to treat hyperglycemia.

## What did the researchers do and find?

- We used 2 independent cohorts of 863 (primary cohort) and 316 (validation cohort) preterm infants born at <30 weeks' gestation to study the association between severe retinopathy of prematurity and 2 markers of glucose exposure between birth and day 21.

- We used a third cohort—a prospective, nationwide population-based cohort of 1,441 preterm infants born at <30 weeks' gestation, representing a large variation of practices regarding neonatal management—to examine the impact of strategies to avoid or reduce hyperglycemia on the risk of severe retinopathy of prematurity.

- Strengthened by multiple sensitivity analyses and external validation, our results support the hypothesis that hyperglycemia is an independent risk factor for severe retinopathy of prematurity and not a mere marker of illness.

- More importantly, we identified thresholds of combined severity and duration of hyperglycemia above which the risk of severe retinopathy increases significantly.

- In the nationwide population-based cohort, the analysis of insulin as a protective factor against severe retinopathy of prematurity found a large but not significant effect after controlling for confounding factors.

## What do these findings mean?

- The results suggest that overall average exposure and duration of hyperglycemia matter more than a single high glucose value when determining risk of retinopathy.

- The thresholds of combined severity and duration of exposure that were identified may help physicians to determine treatment strategies in hyperglycemic premature infants.

- The clinical impact of avoiding exceeding these thresholds to prevent severe retinopathy of prematurity deserves evaluation.

## Introduction

Retinopathy of prematurity (ROP) is a multifactorial sight-threatening disease that remains a challenge in neonatal care, with limited change in incidence in the past 20 years, but increasing rates of survival among at risk infants [1–3]. Prevention by the identification and reduction of risk factors that disrupt normal retinal vascularization at an early stage is likely to be more effective than later treatment of neovascularization [4]. Among potential modifiable risk factors, hyperglycemia has been shown to impair retinal angiogenesis during retinal development in animal models of retinopathy [5,6].

In very preterm infants, the incidence of hyperglycemia is high, reaching 20% to 86%, due to immature insulin production and insulin resistance [7]. A number of epidemiological studies suggested that hyperglycemia was associated with ROP [8–13]. However, most of these studies were based on small retrospective cohorts, and many were conducted in middle-income countries, where babies affected with ROP are bigger and more mature, leaving

uncertainty as to the applicability of their findings in higher income countries. Moreover, not all important confounding factors were evaluated. Accordingly, the authors of a recent meta-analysis concluded that uncertainty remained as to whether hyperglycemia was an independent risk factor for ROP or a mere marker of severity of illness, and called for further studies adjusting for potential confounding factors to clarify this association [14].

Neonatal hyperglycemia is often treated with insulin infusion, but it is still uncertain which concentration of blood glucose could be associated with increased risk of tissue damage, leading to a lack of consensus with regard to the cutoff level to treat hyperglycemia [15,16]. The use of insulin therapy in hyperglycemic preterm infants is itself controversial, since this approach seems to offer little clinical benefit on long-term outcomes while increasing the risk of hypoglycemia [7,17,18].

A randomized controlled trial (RCT) comparing the effects of a restricted versus a liberal approach of hyperglycemia management on the occurrence of severe ROP would be difficult to conduct with an appropriate sample size because of the small numbers of premature infants affected by both severe ROP and hyperglycemia and the multifactorial pathophysiology of ROP. Attempting to reduce severe ROP, which can lead to blindness, is crucial in premature infants. Therefore, using an epidemiological approach to attempt to clarify the link between hyperglycemia and ROP remains legitimate.

We used 2 independent, monocentric cohorts of neonates born at less than 30 weeks' gestation with extensive biological data from institutional databases—a primary and a validation cohort—to analyze the association between exposure to hyperglycemia during the first 21 days of life and severe ROP, in order to identify threshold levels of combined duration and blood glucose concentration that are associated with increased risk of stage 3 or higher ROP. We used EPIPAGE-2 [19], a national population-based prospective cohort study representing a large variation of practices, to examine the impact of strategies to avoid or reduce hyperglycemia on the risk of ROP by studying the association between use of insulin therapy and severe ROP. We hypothesized that exposure to hyperglycemia was independently associated with severe ROP and that insulin therapy, a marker of an attempt to control hyperglycemia, would be associated with less frequent ROP.

## Methods

### Data sources and study participants

We obtained data from 3 sources. The primary and the validation cohorts are 2 independent, monocentric cohorts consisting of all preterm infants born before 30 completed weeks' gestation, consecutively admitted to the neonatal intensive care unit (NICU) of Nantes University Hospital (1 January 2006–31 December 2016, $n$ = 863, primary) [20] or Lyon-HFME University Hospital (1 January 2009–31 December 2017, $n$ = 316, validation) and alive at 36 weeks of postmenstrual age. The third cohort, the EPIPAGE-2 cohort, is a prospective, nationwide population-based cohort of preterm infants born in 2011, with a defined period of recruitment between 28 March 2011 and 31 December 2011 over all regions of France, except one (see [19] for details of recruitment); we restricted the study to infants born at less than 30 weeks' gestation. Data collection and processing of the 3 cohorts was approved by the appropriate ethics committees (Consultative Committee on the Treatment of Information on Personal Health Data for Research Purposes and Committee for the Protection of People Participating in Biomedical Research) and by the National Data Protection Authority (Commission Nationale de l'Informatique et des Libertés, number 17253, 915452, and 911009). Data were anonymized at time of access.

## Main outcome: ROP

Our primary outcome was severe ROP, defined as stage 3 or higher ROP according to the International Classification of Retinopathy of Prematurity [21]. Data regarding maximum stage of ROP in either eye were collected prospectively in all 3 cohorts. In the primary cohort, screening for ROP was conducted using digital retinal imaging exclusively; retinal photographs of all infants diagnosed with ROP were reviewed before analysis, and ROP grading was ascertained by a single ophthalmologist masked to glucose exposure as well as the infants' clinical and biological data. In the validation cohort and the EPIPAGE-2 cohort, screening for ROP was conducted by ophthalmologists using either indirect fundoscopy or digital retinal imaging, both of which are gold standard techniques for ROP screening [22].

## Risk factors

Clinical signs of ROP occur following a multifactorial disruption of retinal angiogenesis; they are not observed before 4 weeks of age in 99% of infants [23]. To take into account this sequence of events, we included in the analysis the values of glycemia that were measured before ROP was diagnosed.

**Glycemia.**  Glycemia data were collected from the prospectively entered hospital biological database in both the primary and validation cohorts. The child's medical record number linked biological and clinical data. For each child, we gathered the number of blood glucose measurements and the highest glycemia value on each day from day 1 to day 21, including both bedside whole blood glucose and laboratory serum glucose measurements. The variation in reported measurements between these techniques is lower than 1 mmol/l [24], and the variance is mostly observed at low glucose concentrations.

**Insulin therapy.**  Data on insulin therapy were collected at 28 days as a binary variable in the EPIPAGE-2 cohort. Infants receiving any insulin therapy during the first 4 weeks of life were classified as exposed.

## Other characteristics of preterm infants

In the 3 cohorts, all data, including demographic data, neonatal morbidities, and biological data were prospectively collected during NICU hospitalization. Gestational age was determined based on the first-trimester ultrasonography. We expressed birth weight as z-scores using the λ-μ-σ method (LMS) from Olsen et al.'s intrauterine growth curves, taking into account sex and gestational age [25]. Glucose, lipid, and protein intakes followed 2005 ESPGHAN recommendations. We measured weight gain during hospitalization by the change in weight z-score from birth to discharge, to represent the adequacy of nutritional intakes [26]. We defined duration of oxygen use as the total number of days during which the infant received supplemental oxygen during any part of the day. We used C-reactive protein and procalcitonin values any time during the first 21 days as a proxy for exposure to sepsis and inflammation.

## Statistical analysis

The preplanned analysis (S1 Text) was hypothesis-driven and did not differ from the final analysis other than in the performance of sensitivity analyses and in the addition of an instrumental variable to clarify the association between insulin treatment and severe ROP, analyses that were undertaken following peer review comments.

We used data from the primary and validation cohorts to analyze the association between blood glucose concentration and severe ROP.

We used 2 markers of blood glucose exposure between birth and day 21, namely maximum value of glycemia ($MaxGly_{1-21}$) and mean of daily maximum values of glycemia ($MeanMaxGly_{1-21}$) to analyze the association between blood glucose concentration and severe ROP. First, we conducted several multiple regression analyses to study the association between each one of these markers and severe ROP before and after adjustment for potential confounders, i.e., the main acknowledged risk factors for ROP (gestational age, birth weight, oxygen therapy, postnatal weight gain, sepsis/inflammation [2,4]). We performed sensitivity analyses in the subgroup of infants born at less than 28 weeks' gestation, which are more at risk of severe ROP, in both the primary and validation cohorts.

To calculate $MeanMaxGly_{1-21}$, missing blood glucose values were imputed once per day using a linear regression model; imputation model variables included glycemic data from the days before and after the periods without glucose measurements, gestational age, and birth weight $z$-score. We generated 50 independent imputed datasets. Because missing data regarding glycemia probably did not occur at random (in NICUs, blood glucose testing is not usually done in stable patients after parenteral nutrition has been discontinued because the risk of dysglycemia becomes very low), we performed sensitivity analyses with imputation of missing glucose data based on different plausible scenarios following van Buuren and Groothuis-Oudshoorn's approach [27] (see S2 Text for details).

The data used to calculate $MaxGly_{1-21}$ were not imputed. Therefore, they were used for the second part of the analyses aiming to identify threshold levels of combined duration and blood glucose concentration that are associated with increased risk of stage 3 or higher ROP. In this second part of the study, we calculated for each infant of the primary cohort the duration (in days) with daily maximum glycemia (MaxGly) value above each concentration of glycemia between 7 and 13 mmol/l by increments of 1 mmol/l. For each concentration of glycemia, we analyzed the association between exposure time and severe ROP using receiver operating characteristic (ROC) curves, and identified the most discriminatory value of exposure time ("optimal cutoff") as the cutoff value with the highest Youden index, a common summary measure of the ROC curve that defines the maximum potential effectiveness of a marker by optimizing both its sensitivity and sensibility [28]. Finally, in the validation cohort, we calculated the sensitivity and specificity of each optimal cutoff value of exposure time above the different concentrations of glycemia to predict severe ROP.

In the third part of the study, we analyzed the association between insulin treatment and severe ROP, using the EPIPAGE-2 cohort, a nationwide cohort, where this association is less prone to depend on each infant's clinical severity due to differences in NICU strategies regarding insulin use.

We applied propensity score weighting to control for observed confounding factors that might influence both group assignment, i.e., exposed and not exposed to insulin therapy [29]. The propensity score was defined as infants' probability of having insulin therapy based on their individual observed covariates. Probability was estimated using a logistic regression model with insulin therapy as the dependent variable in relation to baseline maternal and infant characteristics (see S4 Table for details). We used generalized estimating equations to take into account the center effect. We performed a main analysis on the inverse probability of treatment weighting (IPTW) cohort with adjustment for duration of oxygen exposure, and sensitivity analyses on the overall cohort with and without adjustment for gestational age, birth weight percentile, and duration of oxygen exposure; on the IPTW cohort without adjustment; and on the IPTW cohort including imputed data regarding insulin exposure and ROP status. Imputation of missing data was performed by chained equations using the SAS "MI" procedure. Imputation model variables included exposure to insulin, propensity score variables, and outcome. Binary variables were imputed using logistic regression, and continuous

variables using a linear regression model. We generated 50 independent imputed datasets with 20 iterations each. Estimates were pooled according to Rubin's rule [30]. We performed a sensitivity analysis using an instrumental variable using unit preference regarding insulin use as instrument (see S3 Text for details).

All tests were 2-sided. $p$-Values less than 0.05 were considered significant. All statistical analyses were performed using SAS version 9.4 software (SAS Institute) except for ROC curve analyses. We used MedCalc version 10.2.0.0 software (MedCalc Software) to compare ROC curves and to determine cutoff values.

## Results

### Association between exposure to hyperglycemia and severe ROP in the primary cohort

The characteristics of the infants included in the primary cohort are depicted in Table 1. Among the 863 preterm infants included in the analysis (Fig 1), 38 (4.4%) developed ROP stage 3 or higher, and 54,083 blood glucose measurements were recorded. Each neonate had a

**Table 1. Demographic and clinical characteristics of preterm infants included in the primary, validation, and EPIPAGE-2 cohorts.**

| Characteristic | Primary cohort | Validation cohort | EPIPAGE-2 cohort |
|---|---|---|---|
| Number of infants | 1,121 | 560 | 2,136 |
| Year of birth | 2006–2016 | 2009–2017 | 2011 |
| **Infant characteristics** | | | |
| Male sex—$n$/total $n$ (%) | 586/1,121 (52.3) | 298/558 (53.4) | 1,122/2,136 (52.5) |
| Gestational age at birth—$n$ (%) | | | |
| 23–25 weeks | 189 (16.9) | 122 (21.8) | 396 (15.2) |
| 26–27 weeks | 392 (35.0) | 184 (34.6) | 786 (35.4) |
| 28–29 weeks | 540 (48.2) | 244 (43.6) | 954 (49.4) |
| Median birth weight $z$-score (IQR) | 0.00 (−0.73; 0.59) | −0.23 (−0.96; 0.46) | −0.05 (−0.80;0.54) |
| Cesarean delivery—$n$/total $n$ (%) | 825/1,121 (73.6) | 278/553 (50.3) | 1,289/2,122 (62.8) |
| Exposure to antenatal glucocorticoids—$n$/total $n$ (%) | 828/1,121 (73.9) | 383/522 (73.4) | 1,719/2,095 (82.3) |
| Singleton birth—$n$ (%) | 805 (71.8) | 500 (89.3) | 1,453 (68.2) |
| Apgar score less than 7 at 5 minutes—$n$/total $n$ (%) | 256/1,057 (24.2) | 78/290 (26.9) | 476/1,969 (23.4) |
| Death before 36 weeks of postmenstrual age—$n$/total $n$ (%) | 192/1,121 (17.1) | 168/560 (30.0) | 341/2,136 (14.6) |
| Number of survivors at 36 weeks' postmenstrual age | 929 | 392 | 1,795 |
| **Neonatal outcomes (survivors at 36 weeks' postmenstrual age)** | | | |
| Median discharge weight $z$-score (IQR) | −1.02 (−1.70; −0.36) | −0.54 (−1.14; 0.04) | −0.97 (−1.56; −0.43) |
| Median change in weight $z$-score during neonatal hospitalization (IQR) | −0.91 (−1.44; −0.35) | −0.34 (−0.88; 0.08) | −0.97 (−1.51; −0.42) |
| Severe BPD—$n$/total $n$ (%) | 57/929 (6.1) | 8/392 (2.0) | 236/1,696 (12.8) |
| Median number of weeks with supplemental oxygen (IQR) | 0.4 (0; 23) | 0.3 (0.1; 1.4) | 3.4 (0; 7.3) |
| Number of infants without information regarding ROP status—$n$ (%) | 19 (1.7) | 76 (13.6) | 306 (14.3) |
| Severe ROP (stage $\geq$ 3)—$n$/total $n$ (%) | 41/910 (4.5) | 7/316 (2.2) | 40/1,489 (2.3) |
| Number of infants without glycemia data* | 60 | 0 | — |
| Number of infants without information regarding insulin treatment | — | — | 81 |
| Number of infants included in the analysis | 863 | 316 | 1,441 |

To take into account any differences in the sampling process between children born at 24–26 weeks' and 27–29 weeks' gestation and included in the EPIPAGE-2 cohort [19], results (percentages) were weighted by recruitment period.

*Failure to match the data from the clinical and the biological databases.

BPD, bronchopulmonary dysplasia; IQR, interquartile range; ROP, retinopathy of prematurity.

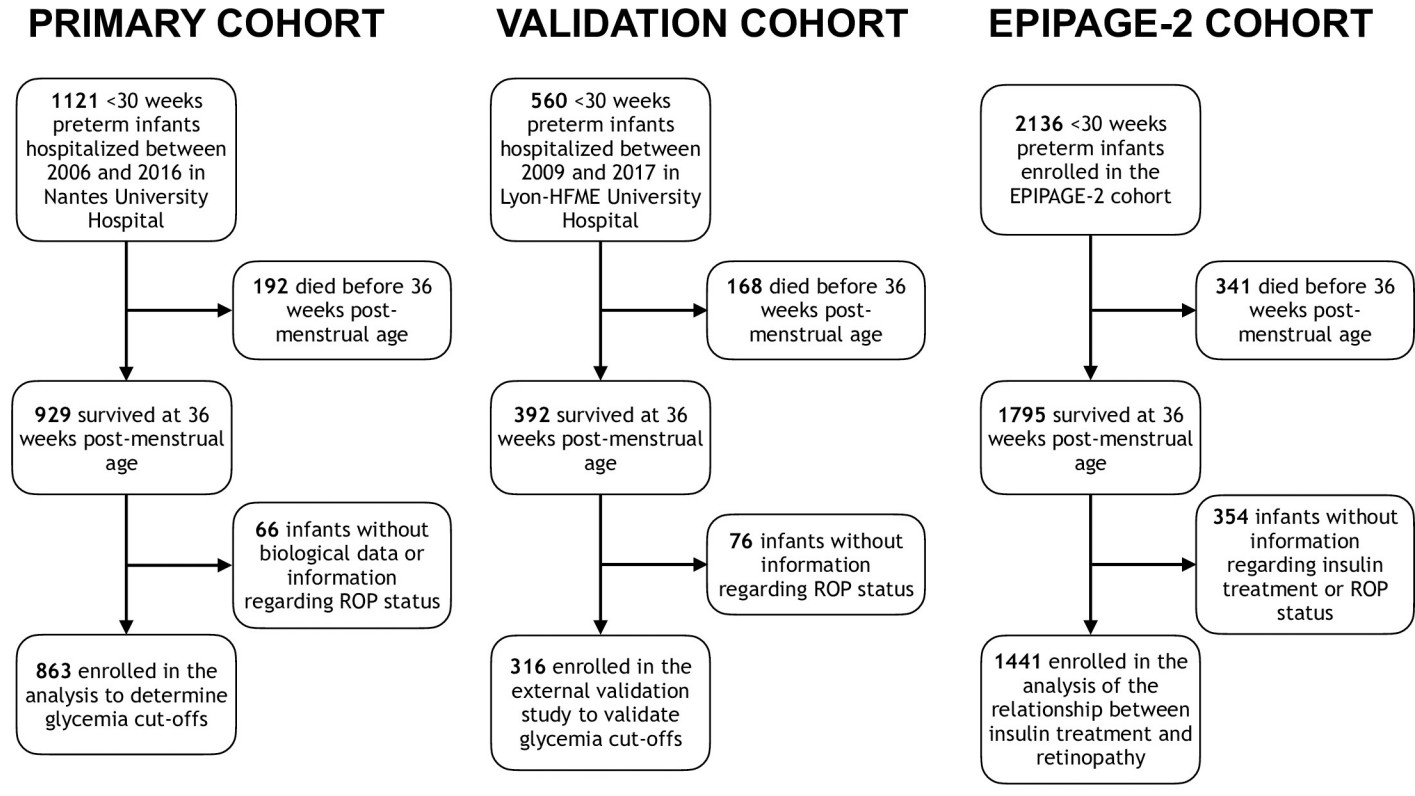

**PRIMARY COHORT**

1121 <30 weeks preterm infants hospitalized between 2006 and 2016 in Nantes University Hospital

→ 192 died before 36 weeks post-menstrual age

929 survived at 36 weeks post-menstrual age

→ 66 infants without biological data or information regarding ROP status

863 enrolled in the analysis to determine glycemia cut-offs

**VALIDATION COHORT**

560 <30 weeks preterm infants hospitalized between 2009 and 2017 in Lyon-HFME University Hospital

→ 168 died before 36 weeks post-menstrual age

392 survived at 36 weeks post-menstrual age

→ 76 infants without information regarding ROP status

316 enrolled in the external validation study to validate glycemia cut-offs

**EPIPAGE-2 COHORT**

2136 <30 weeks preterm infants enrolled in the EPIPAGE-2 cohort

→ 341 died before 36 weeks post-menstrual age

1795 survived at 36 weeks post-menstrual age

→ 354 infants without information regarding insulin treatment or ROP status

1441 enrolled in the analysis of the relationship between insulin treatment and retinopathy

**Fig 1. Flow diagram of the 3 study populations.** ROP, retinopathy of prematurity.

median number of blood glucose measurements of 44 (interquartile range [IQR] 28–78) during the first 21 days of life. $MaxGly_{1-21}$ was higher than 8, 10, 15, and 20 mmol/l in 79.5%, 62.6%, 29.0%, and 12.9% of the 863 infants, respectively. $MaxGly_{1-21}$ was significantly associated with ROP stage 3 or 4 before and after adjustment (Table 2). Sensitivity analyses on a subcohort restricted to the infants born at less than 28 weeks' gestation showed consistent results (S1 Table). Days without glucose measurements composed 31% of all days, mainly in the group of infants in which all glycemic values were <7 mmol/l (36% versus 3% in the group with at least 1 glycemia value $\geq$ 7 mmol/l, $p < 0.001$), and mainly after the first week (41% versus 16% during the first week, $p < 0.001$). Using 50 independent datasets with imputation, we calculated $MeanMaxGly_{1-21}$ for each infant, which was also significantly associated with severe ROP before and after adjustment (Tables 2 and S1).

The association between severe ROP and duration (in days) above different thresholds of MaxGly value was analyzed by calculating the area under the curve (AUC) of the ROC curves for prediction of severe ROP. The AUC for the different thresholds ranged from 0.85 ± 0.035 to 0.87 ± 0.030 and was not significantly different between them (Fig 2A). However, the duration of exposure that was associated with an increased risk of severe ROP varied depending on the severity of hyperglycemia, with optimal cutoffs of 9, 6, 5, 3, 2, 2, and 1 days, for a MaxGly threshold of 7, 8, 9, 10, 11, 12, and 13 mmol/l, respectively. Among the 321 infants with a severe exposure to hyperglycemia (i.e., with at least 1 exposure above 1 of the optimal cutoffs), 35 (10.9%) developed severe retinopathy, compared to only 3 (0.6%) of the 542 infants without severe exposure ($p < 0.001$; Fig 2B). The sensitivity and specificity of each optimal cutoff are presented in Tables 3 and S2.

**Table 2. Association between severe ROP and the maximum value of glycemia between birth and day 21 (MaxGly$_{1-21}$) and the mean of daily maximum values of glycemia between birth and day 21 (MeanMaxGly$_{1-21}$) in the primary and the validation cohorts after adjustment for confounding factors.**

| Analysis | Primary cohort | | | Validation cohort | | |
|---|---|---|---|---|---|---|
| | n | aOR (95% CI) | p-Value | n | aOR (95% CI) | p-Value |
| **Main analysis** | | | | | | |
| **MaxGly$_{1-21}$ (per mmol/l; complete cases)** | | | | | | |
| No adjustment | 863 | 1.21 (1.14–1.27) | <0.001 | 316 | 1.17 (1.05–1.32) | 0.008 |
| Adjustment for gestational age | 863 | 1.13 (1.06–1.21) | <0.001 | 316 | 1.14 (1.00–1.31) | 0.051 |
| Adjustment for birth weight z-score | 838 | 1.21 (1.14–1.28) | <0.001 | 316 | 1.15 (1.01–1.30) | 0.031 |
| Adjustment for postnatal weight gain | 838 | 1.21 (1.14–1.28) | <0.001 | 316 | 1.15 (1.02–1.31) | 0.029 |
| Adjustment for duration of oxygen supplementation | 863 | 1.15 (1.09–1.22) | <0.001 | 316 | 1.14 (1.002–1.30) | 0.047 |
| Adjustment for C-reactive protein | 846 | 1.20 (1.14–1.26) | <0.001 | 314 | 1.18 (1.05–1.33) | 0.005 |
| Adjustment for procalcitonin | 789 | 1.22 (1.15–1.29) | <0.001 | 128 | 1.18 (1.02–1.36) | 0.025 |
| Multiple adjustment including C-reactive protein[a,b] | 821 | 1.09 (1.01–1.18) | 0.026 | | — | — |
| Multiple adjustment including procalcitonin[b,c] | 772 | 1.12 (1.04–1.21) | 0.004 | | — | — |
| **MeanMaxGly$_{1-21}$ (per mmol/l; with multiple imputation)** | | | | | | |
| No adjustment | 863 | 1.70 (1.48–1.94) | <0.001 | 316 | 1.53 (1.20–1.95) | <0.001 |
| Adjustment for gestational age | 863 | 1.36 (1.16–1.60) | <0.001 | 316 | 1.51 (1.11–2.05) | 0.009 |
| Adjustment for birth weight z-score | 863 | 1.72 (1.49–1.97) | <0.001 | 316 | 1.47 (1.14–1.90) | 0.003 |
| Adjustment for postnatal weight gain | 863 | 1.69 (1.47–1.94) | <0.001 | 316 | 1.44 (1.12–1.86) | 0.005 |
| Adjustment for duration of oxygen supplementation | 863 | 1.51 (1.30–1.75) | <0.001 | 316 | 1.46 (1.06–1.96) | 0.012 |
| Adjustment for C-reactive protein | 846 | 1.69 (1.47–1.94) | <0.001 | 314 | 1.59 (1.22–2.06) | <0.001 |
| Adjustment for procalcitonin | 789 | 1.71 (1.48–1.98) | <0.001 | 128 | 1.49 (1.12–1.98) | 0.006 |
| Multiple adjustment including C-reactive protein[a,b] | 846 | 1.18 (0.99–1.41) | 0.070 | | — | — |
| Multiple adjustment including procalcitonin[b,c] | 789 | 1.26 (1.05–1.52) | 0.015 | | — | — |
| **Sensitivity analyses** | | | | | | |
| **MeanMaxGly$_{1-21}$ (per mmol/l, with imputation based on a linear mixed-effects model)** | | | | | | |
| No adjustment | 863 | 1.71 (1.49–1.96) | <0.001 | 316 | 1.50 (1.18–1.90) | <0.001 |
| Adjustment for gestational age | 863 | 1.38 (1.17–1.63) | <0.001 | 316 | 1.48 (1.10–2.00) | 0.011 |
| **MeanMaxGly$_{1-21}$ (per mmol/l, with imputation at random between 4.0 and 6.9 mmol/l)** | | | | | | |
| No adjustment | 863 | 1.75 (1.52–2.00) | <0.001 | 316 | 1.55 (1.21–1.99) | <0.001 |
| Adjustment for gestational age | 863 | 1.42 (1.21–1.66) | <0.001 | 316 | 1.53 (1.12–2.09) | 0.007 |

Confounders were entered in the models as continuous variables.

[a]Adjustment for gestational age, birth weight z-score, postnatal weight gain, duration of oxygen supplementation, and C-reactive protein.

[b]In the validation cohort, full adjustment for all potential confounders in the same model was not performed due to a too small number of cases/potential confounders ratio to estimate regression coefficients reliably.

[c]Adjustment for gestational age, birth weight z-score, postnatal weight gain, duration of oxygen supplementation, and procalcitonin.

aOR, adjusted odds ratio; MaxGly$_{1-21}$, maximum value of glycemia between birth and day 21; MeanMaxGly$_{1-21}$, mean of daily maximum values of glycemia between birth and day 21; ROP, retinopathy of prematurity.

## Association between exposure to hyperglycemia and severe ROP in the validation cohort

Of the 316 infants included in the external validation study (Fig 1; Table 1), 7 developed severe ROP (2.2%); 9,771 blood glucose measurements were recorded. After adjustment for confounding, the association between MeanMaxGly$_{1-21}$, MaxGly$_{1-21}$, and severe retinopathy was confirmed, with effect estimates of similar magnitude (Tables 2 and S1). The specificity values of the optimal cutoffs of duration of exposure for glycemia thresholds between 7 and 13 mmol/l were consistent with those observed in the primary cohort (Table 3). Among the 97

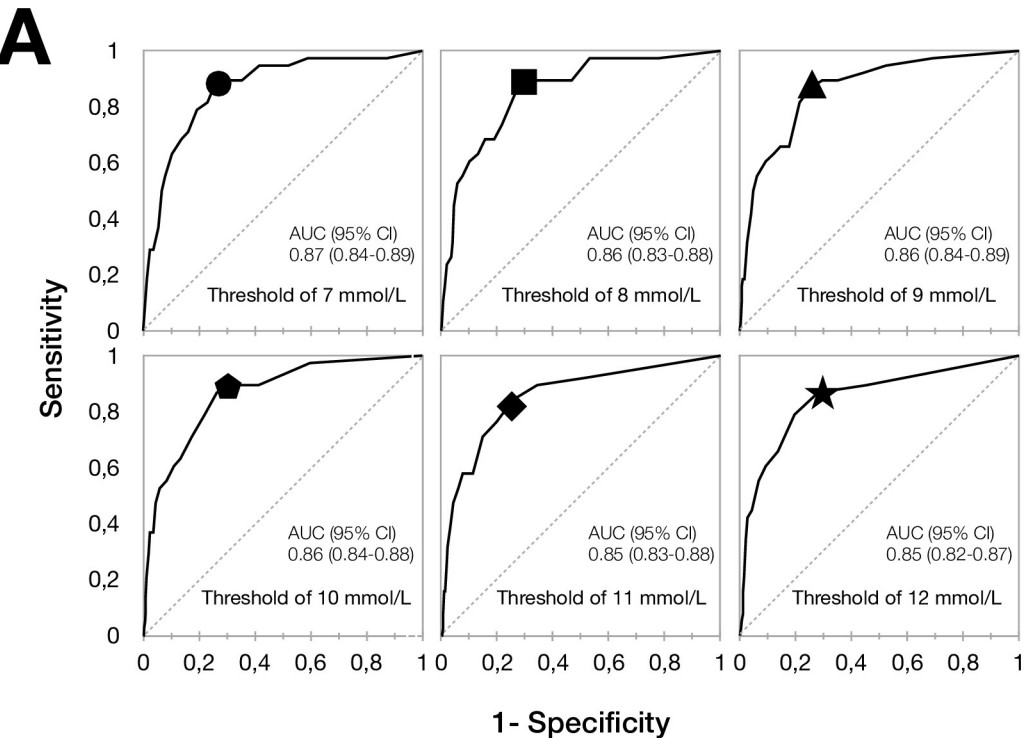

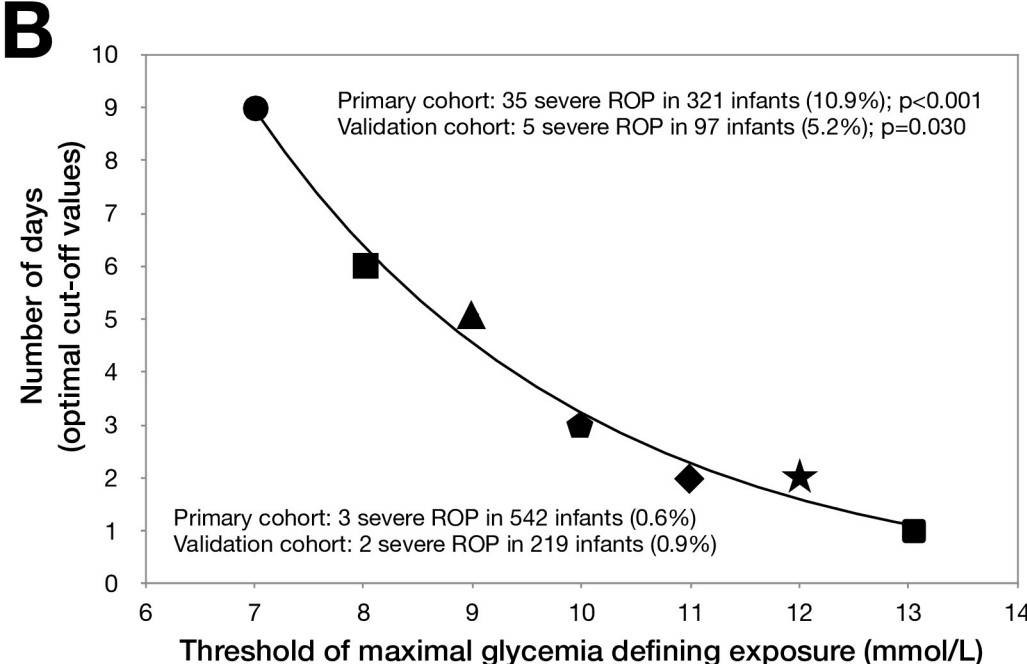

**Fig 2. Thresholds of maximum glycemia and severe retinopathy of prematurity (ROP).** (A) Shown are the receiver operating characteristic (ROC) curves of the duration (in days) with a maximum value of glycemia between birth and day 21 (MaxGly) above 7, 8, 9, 10, 11, and 12 mmol/l as related to severe ROP in the primary cohort. The areas under the curve (AUCs) for the ROC curves ranged from 0.85 (95% CI 0.79 to 0.92) for the duration above 13 mmol/l to 0.87 (95% CI 0.81 to 0.92) for the duration above 7 mmol/l, and were not significantly different ($p = 0.700$). Optimal cutoff values of the number of days above a given threshold of MaxGly were extracted from ROC curve analyses; they are represented by the different symbols. In (B), each optimal cutoff value is plotted versus the corresponding glucose value. Exposure above

these cutoff values was significantly associated with increased risk of severe ROP after adjustment for gestational age in both the primary and validation cohorts.

infants who were exposed to hyperglycemia above 1 of the optimal cutoffs, 5 (5.2%) developed a severe retinopathy, compared to 2 (0.9%) of the 219 infants who were not (*p* = 0.030; Fig 2B).

### Impact of insulin use on the risk of severe ROP in the EPIPAGE-2 cohort

The number of eligible infants from the EPIPAGE-2 cohort was 2,136. Information on exogenous insulin administration was available in 1,714 of the 1,795 infants alive at 36 weeks of postmenstrual age (Fig 1; Tables 1 and S4). Complete results of retinal examinations were available for 1,441 of them, and severe ROP developed in 40 (2.3%); 410 infants received insulin. We calculated a propensity score for insulin use in all neonates to reduce bias in assessing the relationship between insulin treatment and severe ROP. Insulin use was significantly associated with severe ROP in unadjusted analysis (odds ratio [OR] 2.51 [95% CI 1.13–5.58]). After controlling for the observed confounding factors by IPTW, the adjusted OR for stage 3 or higher ROP with insulin was 0.40 (95% CI 0.13–1.24) (S5 Table).

## Discussion

In 2 prospectively recruited cohorts of neonates born at less than 30 weeks' gestation—cohorts that were large relative to similar studies in the past—we found, after multiple adjustment strategies, that high glucose exposure in the first 3 weeks of life was associated with increased risk of severe ROP for each mmol/l increase in the average daily maximum glucose value of the first 21 days. More importantly, we identified the different thresholds of combined level and duration of hyperglycemia above which the risk of severe ROP increases significantly. A

**Table 3. Specificity and sensitivity of the optimal cutoff values of duration of exposure above the corresponding blood glucose concentration (as determined in the primary cohort based on the Youden index) in the primary and validation cohorts.**

| Optimal cutoff value of duration of exposure above glycemic threshold as determined in the primary cohort | Primary cohort | | | | Validation cohort | | | |
|---|---|---|---|---|---|---|---|---|
| | Infants with severe ROP n = 38 | Infants without severe ROP n = 825 | Specificity (95% CI) | Sensitivity (95% CI) | Infants with severe ROP n = 7 | Infants without severe ROP n = 309 | Specificity (95% CI) | Sensitivity (95% CI) |
| (a) More than 9 days with a daily maximum above 7 mmol/l | 34 (89.5) | 217 (26.3) | 0.74 (0.71–0.77) | 0.89 (0.76–0.96) | 5 (71.4) | 74 (23.9) | 0.76 (0.71–0.80) | 0.71 (0.36–0.92) |
| (b) More than 6 days with a daily maximum above 8 mmol/l | 34 (89.5) | 242 (29.3) | 0.71 (0.67–0.74) | 0.89 (0.76–0.96) | 5 (71.4) | 74 (23.9) | 0.76 (0.71–0.80) | 0.71 (0.36–0.92) |
| (c) More than 5 days with a daily maximum above 9 mmol/l | 33 (86.8) | 209 (25.3) | 0.75 (0.72–0.78) | 0.87 (0.73–0.94) | 5 (71.4) | 67 (21.7) | 0.78 (0.74–0.83) | 0.71 (0.36–0.92) |
| (d) More than 3 days with a daily maximum above 10 mmol/l | 33 (86.8) | 218 (25.1) | 0.74 (0.70–0.76) | 0.87 (0.73–0.94) | 5 (71.4) | 69 (22.3) | 0.78 (0.73–0.82) | 0.71 (0.36–0.92) |
| (e) More than 2 days with a daily maximum above 11 mmol/l | 32 (84.2) | 211 (24.3) | 0.74 (0.71–0.77) | 0.84 (0.70–0.93) | 3 (42.1) | 70 (22.7) | 0.77 (0.72–0.81) | 0.57 (0.25–0.84) |
| (f) More than 2 days with a daily maximum above 12 mmol/l | 30 (78.9) | 162 (19.6) | 0.80 (0.77–0.83) | 0.79 (0.64–0.89) | 3 (42.1) | 51 (16.5) | 0.83 (0.79–0.87) | 0.57 (0.25–0.84) |
| (g) More than 1 day with a daily maximum above 13 mmol/l | 31 (81.6) | 171 (20.7) | 0.79 (0.76–0.82) | 0.82 (0.67–0.91) | 4 (57.4) | 55 (21.6) | 0.82 (0.78–0.86) | 0.57 (0.25–0.84) |
| Severe hyperglycemia: (a) or (b) or (c) or (d) or (e) or (f) or (g) | 35 (92.1) | 286 (34.7) | 0.65 (0.62–0.69) | 0.92 (0.79–0.97) | 5 (71.4) | 92 (23.9) | 0.70 (0.65–0.75) | 0.71 (0.36–0.92) |

ROP, retinopathy of prematurity.

third nationwide cohort allowed us to analyze the association between insulin therapy—as a strategy to reduce hyperglycemia—and severe ROP.

ROP is a multifactorial disease characterized by an arrest in physiological retinal angiogenesis and capillary loss [1,2]. The resulting neuronal hypoxia triggers an abnormal angiogenesis response that is responsible for a proliferative vascular disease, which can lead to retinal detachment and blindness. Many factors have been identified in the pathophysiology of ROP development, in particular factors related to exposure to oxygen, oxidative stress, nutritional factors, and growth [1]. Hyperglycemia is commonly associated with many conditions in very preterm infants, including sepsis and intra-uterine growth restriction, that are frequently encountered in infants who later develop ROP and render analysis of the association between hyperglycemia and ROP difficult.

Our study has numerous strengths compared to previous studies investigating the correlation between hyperglycemia and ROP [8–13]. To our knowledge, it is by far the largest study, which allowed us to adjust the results for the main confounding factors in ROP development, including postnatal growth, gestational age, prenatal growth restriction, oxygen exposure, and markers for inflammation. Our investigation also benefits from 2 independent, prospectively collected and representative datasets from a high-income country, including a validation cohort, which increases the robustness of our results. Sensitivity analyses confirmed these results in infants born at less than 28 weeks' gestation, a population more at risk of severe ROP.

As in all observational studies, the main limitations of our study are unmeasured confounding and—despite the large sample size and the very large number of blood glucose samples (more than 63,000)—the lack of measurements of glycemia in some patients beyond the first week of life. Data were prospectively collected; because glucose is a routine, easy-to-obtain surveillance parameter, we believe that missing data regarding glycemia probably concerned the most stable patients weaned from parenteral nutrition, for which common practice in NICUs is to cease regular glucose testing after a few days with normal values, because dysglycemia becomes very unlikely and heel or vein puncture represents one of the most frequent painful procedures endured by neonates [31].

We used multiple statistical approaches to reduce and assess the potential effect of uncontrolled confounding, including multiple adjustment, sensitivity analyses in a subgroup of infants more at risk of severe ROP, and sensitivity analyses to investigate the consequences of different approaches to imputing missing data, all with consistent results. Of note, the data used to identify threshold levels of combined duration and blood glucose concentration associated with increased risk of severe ROP were not imputed.

Another limitation is the overall small number of cases of severe ROP, in keeping with its low incidence in France, similar to other European countries [32], which may have weakened the power of the study of the association between use of insulin therapy and severe ROP in the EPIPAGE-2 cohort. Conversely, a false-positive finding (type 1 error) in the validation study is unlikely despite the small number of severe ROP cases in the validation cohort.

From our analyses, we obtained strong findings to support that hyperglycemia in premature infants is an important risk factor for severe ROP and not a simple marker of severity of illness. Experimental data from animal studies support the idea that hyperglycemia may impose biological changes on the immature retina. Indeed, experimental studies in newborn rodents have shown that early hyperglycemic exposure in ranges similar to those observed in premature infants interferes with normal retinal development and induces both delayed retinal angiogenesis and neuronal loss—similar to ROP [5,6].

While neonatologists are largely in agreement on the need to treat neonatal hyperglycemia in premature infants because it is associated with increased risk of mortality and neurological

morbidity [33–35], the strategies to improve glucose control are still debated. Insulin treatment is usually advocated by experts, to avoid an unwanted restriction of caloric intake. Still, the level of evidence for insulin use is based on a few RCTs showing a beneficial effect of insulin on short-term outcomes only, such as glucose intake and growth [17,36,37]. Very tight glucose control using insulin has been associated with an increased risk of hypoglycemia [17], and prophylactic insulin has been associated with increased mortality [7]. These RCTs did not evaluate severe ROP as a main outcome measure and were not adequately powered to address the issue of ROP prevention via better glucose control. The authors of a small retrospective study suggested that insulin treatment by itself might be a stronger predictor of ROP than hyperglycemia [38]. Unlike this study, we used a propensity score approach in a nationwide cohort to minimize the likelihood of incorrectly attributing the association between hyperglycemia and severe risk of ROP to insulin use (confounding by indication), because the decision to give insulin depends on the infant's clinical state. In the propensity-score-weighted cohort, adjustment for gestational age, birth weight percentile, and duration of oxygen therapy had a large but not significant effect on the association between insulin treatment and severe ROP (unadjusted OR 2.51; adjusted OR 0.40).

The beneficial effect of insulin use to prevent severe ROP development therefore remains unclear. Alternative strategies aiming at improving glycemic control in the first weeks of life by stimulating endogenous insulin secretion, such as early provision of sufficient protein intake [39–41] and early enteral feeding [41,42], as well as preventing hypophosphatemia—which is associated with an increased risk of hyperglycemia [43]—may be interesting to explore, possibly in an integrated approach to prevention.

While controversy also remains as to when to treat hyperglycemia, our study identified threshold levels of combined duration and blood glucose concentration that are significantly associated with an increased risk of severe ROP: >6 days with at least 1 glycemia value > 8 mmol/l, or >3 days with at least 1 glycemia value > 10 mmol/l, or >2 days with at least 1 glycemia value > 11 mmol/l. Being exposed to any of these situations is associated with a 10-fold increased risk of severe ROP. These data suggest that overall average exposure and duration of hyperglycemia matter more than a single high glucose value. The clinical impact of avoiding exceeding these thresholds to prevent ROP deserves further examination.

## Supporting information

**S1 STROBE Checklist. STROBE checklist.**
(DOCX)

**S1 Fig. Sensitivity analysis.** ROC curve analysis of the maximum daily coefficient of variability and of the coefficient of variability of glucose values during the first 21 days of life compared to maximum value of glycemia.
(DOCX)

**S1 Table. Sensitivity analysis: Association between severe ROP and the maximum value of glycemia between birth and day 21 (MaxGly$_{1–21}$) and the mean of daily maximum values of glycemia between birth and day 21 (MeanMaxGly$_{1–21}$) in the primary and validation cohorts after adjustment for potential confounding factors in infants born at less than 28 weeks' gestation (Table A) and in the primary cohort in infants born at less than 27 weeks' gestation (Table B).**
(DOCX)

**S2 Table. Sensitivity analysis: Specificity and sensitivity of optimal cutoff values of duration of exposure above different glycemic thresholds between 7 and 13 mmol/l in the**

primary and validation cohorts in infants born at less than 28 weeks' gestation (Table A) and in the primary cohort in infants born at less than 27 weeks' gestation (Table B).
(DOCX)

**S3 Table. Sensitivity analysis: Association between the maximum value of glycemia between birth and day 21 ($MaxGly_{1-21}$) and a composite outcome of severe ROP or death in the primary cohort before and after adjustment for potential confounding factors.**
(DOCX)

**S4 Table. Characteristics of infants according to exposure to insulin therapy in the EPI-PAGE-2 cohort.**
(DOCX)

**S5 Table. Insulin therapy as a risk factor for severe ROP.**
(DOCX)

**S1 Text. Methods: Preplanned analysis.**
(DOCX)

**S2 Text. Methods: Imputation of missing blood glucose values.**
(DOCX)

**S3 Text. Sensitivity analysis: Insulin therapy as a risk factor for severe ROP—instrumental variable.**
(DOCX)

## Author Contributions

**Conceptualization:** Elsa Kermorvant-Duchemin, Jean-Christophe Rozé.

**Data curation:** Frank Plaisant, Olivier Claris, Pierre-Yves Ancel, Jean-Christophe Rozé.

**Formal analysis:** Elsa Kermorvant-Duchemin, Guylène Le Meur, Laetitia Marchand-Martin, Cyril Flamant, Raphaël Porcher, Alexandre Lapillonne, Olivier Claris, Pierre-Yves Ancel, Jean-Christophe Rozé.

**Funding acquisition:** Pierre-Yves Ancel, Jean-Christophe Rozé.

**Investigation:** Elsa Kermorvant-Duchemin, Guylène Le Meur, Frank Plaisant, Laetitia Marchand-Martin, Cyril Flamant, Raphaël Porcher, Alexandre Lapillonne, Olivier Claris, Pierre-Yves Ancel, Jean-Christophe Rozé.

**Methodology:** Elsa Kermorvant-Duchemin, Laetitia Marchand-Martin, Raphaël Porcher, Pierre-Yves Ancel, Jean-Christophe Rozé.

**Supervision:** Raphaël Porcher, Pierre-Yves Ancel, Jean-Christophe Rozé.

**Validation:** Elsa Kermorvant-Duchemin, Guylène Le Meur, Frank Plaisant, Laetitia Marchand-Martin, Cyril Flamant, Raphaël Porcher, Alexandre Lapillonne, Sylvain Chemtob, Olivier Claris, Pierre-Yves Ancel, Jean-Christophe Rozé.

**Writing – original draft:** Elsa Kermorvant-Duchemin, Jean-Christophe Rozé.

**Writing – review & editing:** Elsa Kermorvant-Duchemin, Guylène Le Meur, Frank Plaisant, Laetitia Marchand-Martin, Cyril Flamant, Raphaël Porcher, Alexandre Lapillonne, Sylvain Chemtob, Olivier Claris, Jean-Christophe Rozé.

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
