## [Decision Letter · Decision Letter 0]

14 Sep 2020

Dear Dr. Kermorvant-Duchemin,

Thank you very much for submitting your manuscript "Thresholds of glycemia, insulin therapy and risk for severe retinopathy in premature infants: A multiple cohort study." (PMEDICINE-D-19-03700) for consideration at PLOS Medicine. 

[LINK]

In light of these reviews, I am afraid that we will not be able to accept the manuscript for publication in the journal in its current form, but we would like to consider a revised version that addresses the reviewers' and editors' comments. Obviously we cannot make any decision about publication until we have seen the revised manuscript and your response, and we plan to seek re-review by one or more of the reviewers. 

We expect to receive your revised manuscript by Oct 05 2020 11:59PM. Please email us (plosmedicine@plos.org) if you have any questions or concerns.

We look forward to receiving your revised manuscript. 

Sincerely,

Emma Veitch, PhD

PLOS Medicine

On behalf of Clare Stone, PhD, Acting Chief Editor, 

PLOS Medicine

plosmedicine.org

*At this stage, we ask that you include a short, non-technical Author Summary of your research to make findings accessible to a wide audience that includes both scientists and non-scientists. The Author Summary should immediately follow the Abstract in your revised manuscript. This text is subject to editorial change and should be distinct from the scientific abstract. Please see our author guidelines for more information: https://journals.plos.org/plosmedicine/s/revising-your-manuscript#loc-author-summary

*Please clarify in the paper if the analytical approach reported here corresponds to one laid out in a prospective protocol or analysis plan? Please state this (either way) early in the Methods section.

*We noted a couple of areas particularly in the Discussion (but may also be in other parts of the paper) where the reporting immediately goes to assuming that the effects observed in this analysis represent causal effects (despite that elsewhere the authors acknowledge the possibility of residual confounding. For example in the discussion (1st sentence) - "we found...that high glucose exposure in the first 3 weeks of life **increased** the risk of severe ROP"; (final para of discussion) - "our study identified thresholds levels of combined duration and blood glucose concentration that increase the risk of severe ROP significantly...". Both might be better stated as "associated with increased risk" for example.

Comments from the reviewers:

Reviewer #1: The authors examined the impact of different levels of hyperglycemia on the risk of severe retinopathy in premature infants. The authors showed, on the basis of three independent cohorts, that high levels of glucose were an independent risk factor for developing retinopathy in premature infants. The study was conducted in a unique study sample, and conducting the study in three independent samples is a major strength. 

I have a number of comments/suggestions that need to be addressed before this manuscript should be considered for publication.

1) Somewhat more details should be given about the study populations. How were the infants included in the study? How was the participation rate? Was written informed consent from the parents available? It is not clear form the text whether the cohort was a routine care database (and thus retrospective) or a prospective cohort study (with inclusion etc etc). 

2) I would recommend to add AUC values to the ROC curves (Figure 2A). 

3) A high proportion of the infants died before enrollment in the analyses. Did the authors considered competing risks in the analyses given that hyperglycemia and mortality could be related as well? 

4) It would have bene of interest not only to look at the mean of peak glucose value, but also to take into account the inter-day variability in glucose levels in relation to ROP. Finding something here would make a stronger argument to frequently measure glycemic levels. 

5) The number of measures of glycemic levels could be related to ROP, as an indication of glycemic variability.

6) Table 3. It would have been informative not to change both the duration and maximum value at the time, which makes the analyses difficult to care to each other. 

7) Page 18. Which confounding factors had most impact on the observation? It is perhaps strange that the effect estimate becomes protective. I am not sure whether propensity scores will exclude the possibility of confounding by indication in this analysis. Can the authors command on this? Did the authors perform a predefined power calculation? Given the low use of insulin in combination with a low prevalence of ROP, it is quite likely you will easily run out of statistical power. 

Reviewer #2: I confine my remarks to statistical aspects of this paper. These were generally fine but I have a couple issues to resolve before I can recommend publication.

First, why divide the stages of ROP into two? I looked up the 5 stages and surely stage V is worse than IV and IV worse than III. So ... I suggest leaving ROP as a 5 level variable and using ordinal logistic regression.

Line 208: Please define the Youden index. Not everyone will know what it is. More importantly, you have to justify its use. Maximizing the Youden index assumes that false positive and false negative conclusions are equally bad. I don't know much about ROP or insulin treatment, but it's not, in general, the case that the two errors are equally bad. This needs justification or modification.

Peter Flom

Reviewer #3: Review of article "Thresholds of glycemia, insulin therapy and risk for severe retinopathy in premature infants: A multiple cohort study."

I have no competing interest on this subject.

Review:

1. This paper does address an important question if hyperglycemia is associated with worsening ROP, a significant important disease affecting extremely preterm infants.

2. I feel that this paper does not advance the field - there are specific concerns in this paper:

a. Line 41 states hyperglycemia causes tissue damage - this is not supported in literature - it may be associated but not causative.

b. The level of hyperglycemia is quite elevated - for example 63% of infants had glucose 180 mg/dL and 29% were greater than 20 mg/dL. 

c. Line 73-5 contrary to what the authors state - the incidence of severe ROP has decreased over the time of this study due to improved control of supplemental oxygen delivery. 

d. The authors do not state the level of C reactive protein and calcitonin that is elevated - IE a surrogate for sepsis and marker of inflammation. They also don't state the timing of and the number of values obtained per infant. Elevation of these can be a late finding in sepsis. 

e. Lines 190-197 are wordy and difficult to understand. 

f. In table 1, the 2 populations ar not comparable in survival, change in weight, and incidence of BPD.

g. Around line 282 - It appears that most of the infants with elevated glucose had significantly elevated values as it appears that most had at least one level > 13 mmol/L (234 mg/dL). 

h. This paper would be enhanced if they focused on infants with gestational age < 27 weeks and infants with gestational age > 26 weeks rarely have severe ROP. Including the older infants in the analysis does not add much and may confound the data evaluation. 

3. This topic is of general interest to neonatologists, pediatricians, and pediatric ophthalmologists.

[LINK]

---

## [Decision Letter · Decision Letter 1]

30 Oct 2020

Dear Dr. Kermorvant-Duchemin,

Thank you very much for re-submitting your manuscript "Thresholds of glycemia, insulin therapy and risk for severe retinopathy in premature infants: A multiple cohort study." (PMEDICINE-D-19-03700R1) for review by PLOS Medicine.

I have discussed the paper with my colleagues and the academic editor and it was also seen again by three reviewers. I am pleased to say that provided the remaining editorial and production issues are dealt with we are planning to accept the paper for publication in the journal.

[LINK]

We look forward to receiving the revised manuscript by Nov 06 2020 11:59PM. 

Sincerely,

Artur Arikainen

Associate Editor

PLOS Medicine

plosmedicine.org

Requests from Editors:

1. Please address any final review comments below.

2. Title: Please amend to: “Thresholds of glycemia, insulin therapy and risk for severe retinopathy in premature infants: A cohort study.”

3. Data Availability Statement: Since the data are available on request, please amend your response to the first question, which currently states “Yes - all data are fully available without restriction”. Please then also describe the reasons for why data are only available on request, eg. patient confidentiality.

4. Abstract: 

a. Please include cohort recruitment dates, and the setting(s). Please remember to include the number of participants in the EPIPAGE-2 cohort.

b. Please include summary participant demographics (age, sex).

c. Please quantify all results with p values and 95% CIs, eg. line 50.

d. Line 61: Please name the main factors that were adjusted for.

e. Line 64: Please replace “subjects” with “patients”.

f. Line 67: Please rephrase as: “In this study, we observed that exposure…”

5. Author summary: 

a. Lines 75 and 79: Please briefly define ‘hyperglycemia’ and ‘retinopathy’ to a lay reader.

b. Line 92: Please clarify the following a bit more: “…representing a large variation of practices,..”.

c. Lines 95-96: Please change “…our results establish that…” to something like “…our results suggest that…”.

d. Line 106: Please clarify that “…when determining risk of retinopathy.”

e. Line 108: Please replace “neonatologists” with “physicians” for clarity.

6. Methods:

a. Please include day and month in cohort recruitment dates.

b. Please clarify whether data were anonymised at time of access, or whether participants provided written informed consent for the use of their data.

c. Please include the relevant prospective analysis plan or protocol with your revised manuscript as a Supporting Information file to be published alongside your study, and cite it in the Methods section, around line 214. A legend for this file should be included at the end of your manuscript.

7. Please report exact p values over 0.001, or p<0.001 otherwise.

8. At various points (e.g., lines 63, 370) you mention the "large" number of participants. Please clarify that this is large relative to similar studies in the past.

9. Lines 370 and 390: Please replace “prospective” with “prospectively-recruited”, or remove altogether – you analyses are nevertheless retrospective.

10. PLOS does not permit "data not shown.” Please remove this claim, or do one of the following:

a) If you are the owner of the data relevant to this claim, please provide the data in accordance with the PLOS data policy, and update your Data Availability Statement as needed.

b) If the data not shown refer to a study from another group that has not been published, please cite personal communication in your manuscript text (it should not be included in the reference section). Please provide the name of the individual, the affiliation, and date of communication. The individual must provide PLOS Medicine written permission to be named for this purpose.

c) For any other circumstance, please contact the journal office ASAP.

11. Line 423: Please rephrase “consensual” to something like: “Neonatologists are largely in agreement…”

12. Reference 31: Please check the DOI for accuracy.

13. Please rename the S1 Appendix to “S1 Checklist”.

---

Comments from Reviewers:

Reviewer #1: the authors addressed my concerns; I don't have any other issues with this manuscript. 

Reviewer #2: The authors have addressed my concerns and I know recommend publication

Peter Flom

Reviewer #3: I appreciate the answers to our questions.

1. You state in the results that 12.9% of infants in the original cohort had a MaxGly1-21 higher than 20 mmol/L. This results in 111 infants with markedly elevated glucose levels. This is a very high number of infants with such elevated glucose levels. Was this high a percentage of markedly elevated glucose levels present in the other studies?

2. I agree with you decision to focus on infants with >stage 2 ROP with respect to reviewers 2 comment.

3. Table 1 states that the incidence of severe BPD was 6.1% in the primary cohort, but the mean duration of oxygen therapy was 0.4. Did you look at the infants who developed severe ROP and their duration of oxygen therapy or duration of mechanical ventilation? Did any infants under go surgery as that has been recognized as a risk factor for the treatment of severe ROP? 

3. You state that the eye exams started at 31 weeks corrected gestation. It is recommended that these screens start at 31 weeks corrected gestation. Thus, infants born at 24 weeks are only 28 weeks corrected at 4 weeks of age - well before the timing of identification of ROP. Please discuss why you stopped at 21 days.

4.DId you describe how many infants in the EPIPAGE cohort received insulin?

5.In the result discussion, you state that the number of infants with severe hyperglycemia above one cut-off level was 97. However, in table 3, 5 infants with severe ROP and 74 infants without ROP had severely elevated glucose levels. Please explain this discrepancy.

[LINK]

---

## [Editor Report · Decision Letter 2]

19 Nov 2020

Dear Pr Kermorvant-Duchemin, 

On behalf of my colleagues and the academic editor, Dr. Lars Åke Persson, I am delighted to inform you that your manuscript entitled "Thresholds of glycemia, insulin therapy and risk for severe retinopathy in premature infants: A cohort study." (PMEDICINE-D-19-03700R2) has been accepted for publication in PLOS Medicine. 

PRODUCTION PROCESS

Before publication you will see the copyedited word document (within 5 business days) and a PDF proof shortly after that. The copyeditor will be in touch shortly before sending you the copyedited Word document. We will make some revisions at copyediting stage to conform to our general style, and for clarification. When you receive this version you should check and revise it very carefully, including figures, tables, references, and supporting information, because corrections at the next stage (proofs) will be strictly limited to (1) errors in author names or affiliations, (2) errors of scientific fact that would cause misunderstandings to readers, and (3) printer's (introduced) errors. Please return the copyedited file within 2 business days in order to ensure timely delivery of the PDF proof. 

If you are likely to be away when either this document or the proof is sent, please ensure we have contact information of a second person, as we will need you to respond quickly at each point. Given the disruptions resulting from the ongoing COVID-19 pandemic, there may be delays in the production process. We apologise in advance for any inconvenience caused and will do our best to minimize impact as far as possible.

PRESS

PROFILE INFORMATION

Thank you again for submitting the manuscript to PLOS Medicine. We look forward to publishing it. 

Best wishes, 

Artur Arikainen, 

Senior Editor 

PLOS Medicine

plosmedicine.org